# Effects of Daily Lifestyle Habits on Non-Neurogenic Orthostatic Hypotension in Older Adults in South Korea: A Cross-Sectional Study

**DOI:** 10.3390/healthcare13060674

**Published:** 2025-03-19

**Authors:** Nahyun Kim, Hye-Kyung Oh

**Affiliations:** 1College of Nursing, Keimyung University, Daegu 42601, Republic of Korea; drkim@kmu.ac.kr; 2College of Nursing, Daegu University, Daegu 42400, Republic of Korea

**Keywords:** orthostatic hypotension, lifestyle habits, older adults, autonomic functions

## Abstract

**Background:** Orthostatic hypotension (OH) is a chronic, debilitating condition common in older adults. This study examined the effects of daily lifestyle habits on non-neurogenic OH in older adults in South Korea. We further compared the effects of daily lifestyle habits on OH to those of the autonomic nervous system (ANS) function. **Methods:** In a cross-sectional study, 217 community-dwelling older adults aged ≥65 years were recruited using the convenience sampling method. Data were collected using two questionnaires to assess OH and daily lifestyle habits: OH was measured by Orthostatic Grading Scale (OGS) and lifestyle habits included nutrition, sleep, physical activity, and psychological status (stress and depression levels). Plasma catecholamines (epinephrine and norepinephrine) were measured to indicate the ANS function. The data were analyzed using *t*-tests, Pearson’s correlation coefficients, and multiple linear regression analysis. **Results:** Significant factors related to OGA score included nutritional status (B = −0.20, *p* ≤ 0.040), poorer sleep quality (B = 0.15, *p* = 0.005), physical activity (B = −0.01, *p* = 0.032), stress (B = 0.04, *p* = 0.001), and depression (B = 0.23, *p* = 0.001). These together explained 40.5% of the variance in OH. However, no significant association was found between catecholamines and OGS score. **Conclusions:** These results suggest that lifestyle habits are important factors, while ANS function may be less associated with non-neurogenic OH. Thus, preventive and non-pharmacological interventions for decreasing OH symptoms should focus on maintaining healthy lifestyle habits in older adults.

## 1. Introduction

Orthostatic hypotension (OH) is a chronic, debilitating condition that is difficult to treat in older adults [1]. OH is defined as a drop in systolic blood pressure of at least 20 mmHg and in diastolic blood pressure of at least 10 mmHg within 3 min of standing up [2]. OH is common in adults aged ≥ 65 years, and the prevalence of OH increases exponentially with advancing age, affecting over a quarter of older adults in the general population and over a third of geriatric outpatients [3]. The main reason for the high OH prevalence in older adults is probably altered baroreceptor sensitivity [4]. In addition, aging is related to generalized neural loss, a reduced number of beta receptors with lowered functionality, and a weaker response to catecholamines, resulting in alterations in autonomic nervous system (ANS) function, all of which may contribute to OH development [5]. Increased arterial wall stiffness, reduced left ventricular compliance, dehydration due to an impaired thirst response, sarcopenia, and polypharmacy also seem to account for OH in a large proportion of cases involving older adults [6,7,8]. Therefore, it can be assumed that aging itself can increase the incidence of OH regardless of any coexisting diseases. Manifestations of OH resulting from cerebral hypo-perfusion include lightheadedness, dizziness, fainting, vision changes, syncope, and cognitive impairment [9], making OH one of the leading causes of falls and fractures in older adults [10].

Regarding its pathophysiological basis, OH may be divided into two categories: neurogenic and non-neurogenic disorders and conditions [10]. Neurogenic OH is a highly prevalent feature of autonomic failure due to central and peripheral neurodegenerative diseases [11,12]. In contrast, non-neurogenic OH is more frequent in the general population [13] and can be determined by factors such as aging, drug or substance use, cardiac impairment, fluid and electrolyte deficits, deconditioning, or abrupt postural changes other than autonomic failure [10,14]. The presence of OH, regardless of neurogenic or non-neurogenic mechanism, impairs quality of life [15,16] and can contribute to a greater risk of incident comorbid diseases and all-cause mortality [17,18]. Thus, OH is becoming a significant public health burden [19].

Management of OH is aimed at alleviating symptoms and improving quality of life rather than normalizing blood pressure [12,20]. Previous studies have reported the effects of both pharmacological and non-pharmacological approaches on the alleviation of OH symptoms. Regarding the pharmacological approach, medications for increasing sympathetic tone, blood volume, and peripheral vascular resistance are commonly used for OH treatment; however, their long-term use is usually prohibited due to their side effects, including hypertension and organ damage [21,22,23]. Non-pharmacological interventions, such as an increase in water and sodium intake, head elevation while sleeping, administration of elastic abdominal binders, compressive stockings, and physical counter maneuvers, have been recommended as first-line treatments [15,22,24,25,26,27]. However, non-pharmacological interventions reported in the literature are often unsuitable for older adults with multiple comorbidities or contraindications, and compliance and tolerability may also be limited in this population [28]. Considering these concepts, safer, easier to adopt, more fundamental, and more integrated approaches into daily life should be alternatively considered, such as daily lifestyle modifications for older adults with OH [18].

Recently, evidence for successful interventions using key lifestyle habits (nutrition, sleep, physical activity, and stress management) to manage and control health has been increasing [29]. Depression or depressive symptoms are also included as a lifestyle indicator [30] and are closely related to various cardiovascular disorders and OH in older adults [31]. However, not all results have been consistent; for example, nutritional status and depression in older adults did not correlate with OH [2,32]. Moreover, although the incidence of non-neurogenic OH is higher than that of neurogenic OH [13], most research has focused on neurogenic OH. Additionally, many studies have reported an association between each lifestyle habit and OH; however, there are few studies based on substantial evidence for older adults. Therefore, more studies based on empirical data concerning the effect of daily lifestyle habits on OH are needed.

This study examined the effects of daily lifestyle habits on OH. We focused on non-neurogenic OH associated with aging and excluded neurogenic OH due to underlying autonomic dysfunction. In addition, we compared the effects of these factors on OH with those of basal levels of ANS function in older adults. We hypothesized that daily lifestyle habits would be significantly associated with OH in older adults and that this association would be stronger than the association with ANS function.

## 2. Materials and Methods

### 2.1. Study Design

This study used a cross-sectional design.

### 2.2. Participants and Procedures

The inclusion criteria were adults (a) aged ≥ 65 years, (b) capable of independently engaging in activities of daily living, (c) capable of oral communication, (d) who scored ≥ 24 on the Korean version of the Mini-Mental State Examination (MMSE). Individuals were excluded if they (a) had a history of disorders causing neurogenic OH including chronic renal failure, Parkinson’s disease or other neurodegenerative disease; (b) had diagnosed autonomic neuropathy caused by diabetes mellitus or other diseases; and (c) experienced dizziness not associated with orthostasis, such as vestibular dysfunction or anemia. In addition, we also excluded participants if they (d) had a recent history of taking medications (e.g., antidepressants, diuretics, α-adrenergic receptor blockers, β-adrenergic receptor blockers, or sedatives) that might cause OH [33]. The researchers reviewed the list of medications provided by the participants, and when we were unsure whether a medication could cause OH, we consulted a pharmacist for clarification of the medication’s effects.

To obtain a medium effect size of 0.15, significance level of 0.05, and power of 0.94 using G*power 3.1.9.2 (University of Düsseldorf, Düsseldorf, Germany), the minimum sample size was 172. Convenience samples of 288 community-dwelling older adults were recruited from three community centers in a city in Korea. After screening of older adults for eligibility, a total of 217 individuals were included in the study. The procedure for selecting participants is shown in Figure 1. Each participant completed a set of study questionnaires, and venous blood samples were drawn in a private room in the early morning (08:00 to 10:00 a.m.) after 12 h of fasting. They abstained from caffeine-containing foods and beverages for 24 h prior and had sufficient sleep the night before data collection. Further description about the participants’ inclusion and exclusion criteria and a recruitment procedure are provided elsewhere [16,34].

### 2.3. Measurements

The seven items concerning participant personal characteristics and medical information were determined by a researcher based on previous studies and focused on sex, age, education, blood levels of epinephrine and norepinephrine, MMSE score, and comorbidities.

The Orthostatic Grading Scale (OGS) was used to assess OH levels. In Western countries, the OGS is generally considered a reliable and valid measure for assessing orthostatic symptoms, including orthostatic dizziness [35]. The OGS is a 5-item questionnaire designed to measure the severity of OH symptoms and consists of questions frequently asked to screen for OH symptoms in clinical settings [36]. It has been translated into Korean, and its reliability and validity have been established [37]. Higher scores indicate greater severity of OH. It is widely used in screening for orthostatic symptoms or OH because of its convenience and low cost [38].

Nutritional status was measured using the Mini Nutritional Assessment Short Form (MNA-SF) [39]. The sum of the MNA-SF scores distinguished older adults with (a) normal nutritional status (12–14 points), (b) risk of malnutrition (8–11 points), and (c) malnutrition (0–7 points).

Sleep quality was measured using the Korean version of the Pittsburgh Sleep Quality Index (PSQI-K) [40]. This index assesses sleep quality and disturbances during an interval of 1 month. It consists of 19 items that generate seven component scores: subjective sleep quality, latency, duration, disturbances, use of sleeping medication, habitual sleep efficiency, and daytime dysfunction. The sum of the scores for these seven components yields global sleep quality score. The global score of the PSQI-K ranges from 0 to 21, with higher scores indicating poorer sleep quality. Cronbach’s α of the Korean version of the PSQI-K was 0.69 [40].

To assess physical activity levels, we used the Korean version of the Physical Activity Scale for the Elderly (PASE-K) [39]. It is composed of 10 items that measure three domains of activity (leisure, household, and work-related activities) in terms of frequency during the previous 7 days. The test-retest reliability of the PASE-K was 0.94 [41].

We measured the level of perceived stress using the geriatric stress scale modified by Lee and Kim [42], which was developed by Kim and Kang [43] based on the Family Inventory of Life Events and Changes (FILE) suggested by McCubbin et al. [44]. The FILE consists of 20 items in four sub-domains (health problems, economic problems, family problems, and living environment) and is scored using a 5-point Likert-type scale ranging from “not at all” (1 point) to “very stressful” (5 points). The total score ranges from 20 to 100, with a higher score indicating a higher level of stress perception among older adults. The reliability of the FILE (Cronbach’s α) was 0.92 in the study by Lee and Kim [42] and 0.88 in this study.

The Korean version of the Geriatric Depression Scale-Short Form (GDS-SF) has been used to measure depression levels in older adults [45]. In this study, GDS-SF scores of ≥8 indicated possible depression [46] (Cronbach’s α = 0.88).

To measure autonomic function, blood samples collected via peripheral catheters were assayed for epinephrine and norepinephrine levels. Each participant was instructed to lie quietly for 20 min before blood sampling. About 3 mL of venous blood sample was collected using a heparin anticoagulation vacuum tube, centrifuged at 3000 rpm for 10 min, the plasma separated into polyethylene tubes, which was then stored in a freezer at −20 °C until analysis. Extractions were conducted in 1.5 mL Eppendorf tubes using 50 mg of alumina, 1 mL plasma, 125 uL of 4 M HClO4, 30 uL of 10 mM Sodium metabisulfite and 1 ng/mL of dihydroxybenzylamine (DHBA) as an internal standard. After adding 500 uL of 3 M tris buffer, the resulting mixture was vortex-mixed for 10 min, then centrifuged at 500× *g* for 5 min and the supernatant discarded. This was washed three times with 1 mL water to remove any interfering substances and nitrogen was evaporated to completely remove the water in the alumina. The catecholamines were eluted by vortexing with 200 μL 0.1 M HClO4 for 15 s, followed by centrifugation and transfer of the supernatant into vials for the high-performance liquid chromatography (HPLC) analysis (Agilent 1200 series, Agilent Technology, Santa Clara, CA, USA).

### 2.4. Statistics

All collected data were analyzed using SPSS Statistics 22.0 (IBM Corp., Armonk, NY, USA). We described the participant demographic characteristics, medical information, OH, nutritional and sleep states, physical activity, stress, depression, and the levels of epinephrine and norepinephrine in blood using real numbers, percentages, means, and standard deviations. We analyzed the differences in participant OH according to their general characteristics and medical information using an independent *t*-test. Correlations among major variables were analyzed using Pearson’s correlation coefficients. The factors influencing OH were analyzed using multiple linear regression analyses.

### 2.5. Ethics

This study was approved by the Institutional Review Board of K University (IRB NO. 40525-201609-HR-92-01). Participants were selected based on their willingness to voluntarily participate in the study. Informed consent was obtained from all the participants.

## 3. Results

### 3.1. General Characteristics and Medical Information

Table 1 presents the general characteristics and medical information of the 217 participants who were mostly female (69.1%). The largest proportion belonged to the group aged 75–84 years (49.8%). Regarding the education level, 43.3% of participants had completed primary school or below and 56.7% had an above primary school level. The mean epinephrine and norepinephrine blood scores (pg/mL) were 50.95 ± 20.32 and 423.47 ± 200.35, respectively. The mean MMSE score was 26.83 ± 3.05. In terms of comorbidities, 53.5% had hypertension history, 25.3% had diabetes, 6.5% had a history of stroke, and 8.3% had heart disease.

### 3.2. Daily Lifestyle Habits and OH

In terms of lifestyle, the mean scores for nutritional status, sleep quality, and physical activity were 12.15 ± 2.15, 7.75 ± 61.49, and 85.36 ± 97.09, respectively. The mean scores for stress, depression, and OH were 38.81 ± 14.74, 5.60 ± 3.38, and 3.34 ± 3.49, respectively (Table 2).

### 3.3. OH According to Characteristics

We found significant differences in OH according to age (t = 10.37, *p* = 0.001), educational level (t = 21.56, *p* < 0.001), and hypertension (t = 12.56, *p* < 0.001) (Table 3).

### 3.4. Relationships Among Main Variables

The correlations between the major variables are listed in Table 4. Nutritional status was significantly negatively correlated with poorer sleep quality (r = −0.30, *p* < 0.001), stress (r = −0.41, *p* < 0.001), depression (r = −0.29, *p* < 0.001), and OH (r = −0.36, *p* < 0.001) and significantly positively correlated with physical activity (r = 0.15, *p* = 0.03). Poorer sleep quality was negatively correlated with physical activity (r = −0.31, *p* < 0.001) and positively correlated with stress (r = 0.49, *p* < 0.001), depression (r = 0.40, *p* < 0.001), norepinephrine level (r = 0.16, *p* = 0.017), and OH (r = 0.49, *p* < 0.001). Physical activity was negatively correlated with stress (r = −0.28, *p* < 0.001), depression (r = −0.31, *p* < 0.001), norepinephrine level (r = −0.18, *p* = 0.007), and OH (r = −0.39, *p* < 0.001). Stress was significantly and positively correlated with OH (r = 0.50, *p* < 0.001). Depression was negatively correlated with epinephrine level (r = −0.17, *p* = 0.013) and positively correlated with stress (r = 0.48, *p* < 0.001) and OH (r = 0.48, *p* < 0.001). Epinephrine level was positively correlated with norepinephrine level (r = 0.36, *p* < 0.001).

### 3.5. Factors Influencing OH Level

Multiple regression analyses were conducted to identify the factors independently related to OH. Table 5 presents the results of the regression analyses, which were conducted using a total of predictor variables: MMSE, nutrition statuses, sleep quality, physical activities, stress, depression, epinephrine level, and norepinephrine level. Age and years of education were used as the control variables. The regression model related to OH was found to be significant (F = 17.19, *p* ≤ 0.001), and the adjusted coefficient of determination (Adj R^2^), which indicates the explanatory power of the model, was 0.405. Significant factors related to OH included nutritional status (B = −0.20, *p* ≤ 0.040), poorer sleep quality (B = 0.15, *p* = 0.005), physical activity (B = −0.01, *p* = 0.032), stress (B = 0.04, *p* = 0.001), and depression (B = 0.23, *p* = 0.001). These together explained 40.5% of the variance in OH.

## 4. Discussion

This study was conducted to investigate the effects of lifestyle habits and catecholamines on OH in community-dwelling older adults. We found that lifestyle habits significantly affected OH, and the explanatory power was 40.5%. In contrast, catecholamines, which reflect autonomic nerve function, did not appear to affect OH.

In previous studies, non-pharmacological approaches such as changing positions gradually, drinking water with salts, and elevating the head of the bed were defined as lifestyle modifications [9,25]; however, this study differs from previous studies by successfully confirming the effect of overall lifestyle habits such as nutritional status, sleep, physical activity, and psychological status on OH. In addition, a previous study has shown that OH is associated with an attenuated orthostatic increase in plasma norepinephrine levels [47], while in this study, the epinephrine and norepinephrine plasma levels were not significant factors for non-neurogenic OH. Unlike neurogenic OH, which is associated with central or peripheral autonomic disorders, non-neurogenic OH may result from causes such as aging, adverse medication effects, substance use, cardiac impairment (particularly congestive heart failure), fluid and electrolyte deficits, deconditioning, sarcopenia, or abrupt postural changes other than autonomic failure [8,9,13,48,49]. Therefore, the reason that catecholamines were not associated with orthostatic hypotension symptoms in this study seems to be because the participants of this study had non-neurogenic OH, which is known to have little direct relationship with autonomic dysfunction. However, since non-neurogenic OH is also a phenomenon caused by an insufficient compensatory mechanism of ANS [5], it cannot be ruled out that the null association of plasma catecholamines and OH is due to the adaptation of ANS in this study. ANS has an important role in regulating blood pressure via autonomic vasomotor nerves and circulating catecholamines [1], but persistent stimuli may ultimately result in a reduction of plasma catecholamine responses due to receptor down-regulation reflecting adaptation [50]. Therefore, it is necessary to confirm the relationship between non-neurogenic OH and autonomic nerve function in future studies.

Regarding the factors influencing daily lifestyle habits on OH, this study showed that older adults had a normal nutritional status. The result is similar to that reported by Kim et al. [51] with older adults (aged 70–84 years) in Korea, which indicated that 85.7% of the participants reported normal nutritional status. Kocyigit et al. [52] reported that optimization of nutritional status may improve global cognition and gait balance functions and prevent falls in older people with OH. Nutrition helps reduce the risk of chronic diseases, including OH, leading to improved overall health and well-being. Therefore, nutritional status is thought to be a crucial factor that influences OH prevalence in geriatric patients. In particular, older adults are prone to sarcopenia, and it is critical to maintain muscle mass and strength [8].

The quality of sleep of the participants of this study was poor, similar to the results of other studies targeting Koreans [53]. Poor sleep quality was found to also affect OH; this finding is consistent with that of a previous study [54]. Asensio et al. [55] also showed that sleep dysfunction was worse in participants with OH, although the difference did not reach statistical significance. Improved sleep quality may be necessary to recover cardiac function and ANS. Sleep quality plays a vital role in maintaining good health and in reducing the prevalence of OH.

In this study, physical activity was directly associated with OH. Figueroa et al. [1] mentioned that patients with OH should avoid inactivity and consider a gentle exercise program because mild physical activity improves orthostatic tolerance by reducing venous pooling and increasing plasma volume [56]. In addition, the increased prevalence of OH among older adults may be related to sarcopenia, as reduced lower extremity muscle contractility decreases venous return and makes it difficult to maintain balance [8]. According to a recent systematic review, all types of physical activity showed a concomitant increase in muscle strength together with the ability to balance in older adults [57]. This review further showed that inactivity, compared to some types of physical activity, has a greater impact on balance mechanisms in older adults. Therefore, it is necessary to include not only physical activity but also inactivity when assessing or establishing preventive strategies for OH.

Notably, psychological status (stress and depression) is also associated with OH. Psychological stress increases blood pressure by stimulating the sympathetic nervous system, but in this study, stress was analyzed as a factor affecting OH. In terms of possible mechanisms, stress typically stimulates the cardiovascular system, but chronic stress can reduce baroreflex activity and cardiocirculatory variability. High stress levels may have affected OH by disrupting the baroreflex function during orthostasis [58]. Physiologically, the baroreflex is responsible for maintaining the blood pressure in a normal range. Another plausible mechanism for high blood pressure due to stress is that the risk of OH occurrence is rather high because elevated blood pressure has a greater drop during orthostasis [59]. This phenomenon occurs even when not receiving anti-hypertensive treatment, and for this reason, the prevalence of OH is higher in hypertensive patients than in those with controlled blood pressure [60]. However, since no study has directly reported the effect of psychological stress on OH, further research is needed.

Regarding the association of depression with OH, a majority of studies have shown that depression or depressive symptoms were subsequently followed by OH due to limitations in daily life and lack of social activity because of OH [61]. However, some studies reported that depression could suppress sympathetic activity, contributing to dizziness, OH, or low blood pressure [62,63,64]. Similarly, our results have also shown that depression is a critical factor influencing OH, suggesting a new perspective on the association between depression and OH. The significance of our study is the identification of a psychological status for OH. Thus, psychological factors, such as stress and depression, should be considered important in the treatment of OH.

Taken together, ours and other results suggest that lifestyle habits have a clinically meaningful impact on OH. Many previous studies have focused on pharmacological interventions for OH treatment. However, lifestyle is a preventable and modifiable factor that should be checked and improved in older adults with OH. Figueroa et al. [1] showed that patient education and non-pharmacological strategies alone could be effective for those with mild OH. Improving daily lifestyle habits is an effective approach before OH symptoms worsen. Since daily lifestyle habit changes are highly cost-effective approaches, efforts to change lifestyle can decrease the prevalence of OH at the individual level, and a consistent institutional approach should be prepared. Freeman et al. [65] reported that many patients are asymptomatic despite substantial systolic blood pressure decreases and low orthostatic blood pressure. Therefore, active surveillance and preventive approaches for older adults are becoming increasingly important.

To our knowledge, this study contributes to the literature because this was the first study to demonstrate the promising effects of daily lifestyle habits on OH in older adults. Despite all the efforts made in this study, there are still some limitations. First, the cross-sectional study design limited the possibility of establishing a causal relationship among the main variables. Further longitudinal studies are required to confirm these findings. Second, OH was measured through self-reporting, and further research is necessary to objectively measure OH. Third, to focus on non-neurogenic OH, we excluded elderly people at risk of neurogenic OH during data collection, but they were not medically diagnosed with neurogenic or non-neurogenic OH. Lastly, it should be also addressed that some of the participants had comorbidities known to be associated with OH such as diabetes, stroke, or heart disease, which are prevalent in this age group. Although we included only participants with well-controlled comorbid conditions in our study, caution is needed when interpreting the results given the known association between OH and these comorbidities [10,49].

## 5. Conclusions

This descriptive cross-sectional study identified whether daily lifestyle habits and catecholamines affect non-neurogenic OH in older adults in South Korea. As a result of the multiple regression analysis, lifestyle habits, such as nutritional status, sleep quality, physical activity, and psychological status were demonstrated to be important factors, while ANS function had no effect on non-neurogenic OH. Thus, preventive and non-pharmacological interventions for decreasing OH symptoms should primarily focus on maintaining healthy lifestyle habits in community-dwelling older adults.

## Figures and Tables

**Figure 1 healthcare-13-00674-f001:**
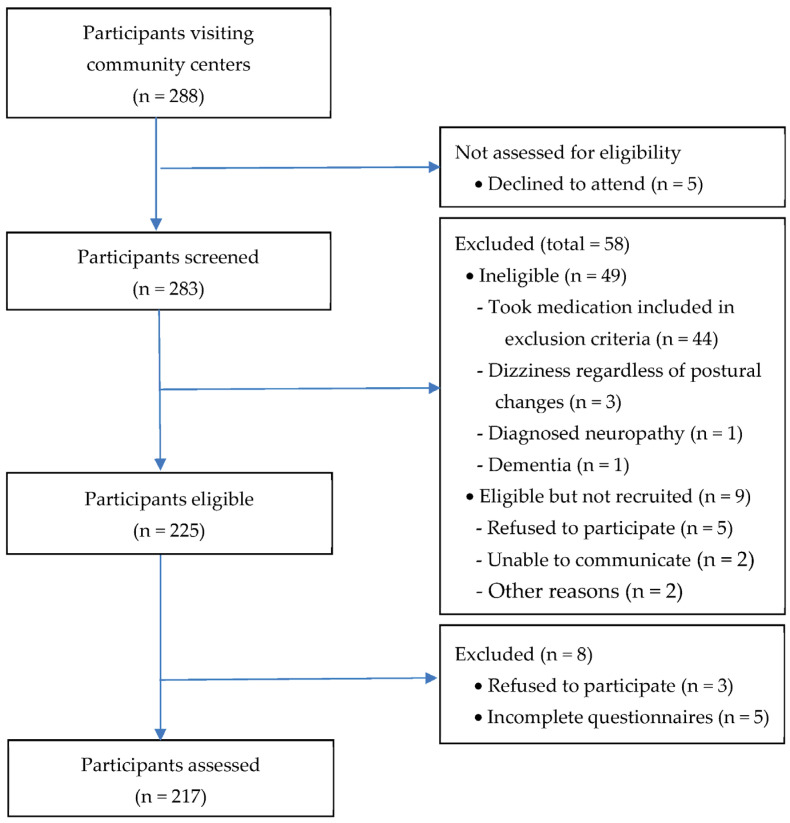
Flow chart depicting selection process for study population.

**Table 1 healthcare-13-00674-t001:** Participant general characteristics and medical information (N = 217).

Variables	Categories/Normal Range	Frequency (%)	Mean ± SD *
Sex	Male	67 (30.9)	
	Female	150 (69.1)	
Age (years)	65~74	96 (44.2)	
	75~84	108 (49.8)	
	Over 85	13 (6.0)	
Education	Primary school or below	94 (43.3)	
	Above primary school	123 (56.7)	
Epinephrine (pg/mL)	0 to 140	-	50.95 ± 20.32
Norepinephrine (pg/mL)	70 to 1700	-	423.47 ± 200.35
MMSE scores (points) *	24~30	-	26.83 ± 3.05
Comorbidities			
Hypertension	Yes	101 (53.5)	
	No	116 (46.5)	
Diabetes	Yes	55 (25.3)	
	No	162 (74.7)	
Stroke history	Yes	14 (6.5)	
	No	203 (93.5)	
Heart disease	Yes	18 (8.3)	
	No	199 (91.7)	

* MMSE = Mini-Mental State Examination, SD = Standard deviation.

**Table 2 healthcare-13-00674-t002:** Daily lifestyle habits and orthostatic hypotension.

Variables		Mean ± SD *	Min	Max
Daily lifestyle habits	Nutritional status	12.15 ± 2.15	4	14
	Sleep quality	7.75 ± 61.49	1	18
	Physical activity	85.36 ± 97.09	5	331
	Depression	5.60 ± 3.38	0	15
	Stress	38.81 ± 14.74	20	91
Orthostatic hypotension		3.34 ± 3.49	0	17

* SD = Standard deviation.

**Table 3 healthcare-13-00674-t003:** Differences in orthostatic hypotension according to variables (N = 217).

Variables	Categories	Mean ± SD *	t (*p*)
Sex	Male	2.73 ± 3.33	2.99 (0.085)
	Female	3.61 ± 3.53	
Age (years)	Under 75 years	2.63 ± 2.95	10.37 (0.001)
	76 years or older	4.13 ± 3.86	
Education	Primary school or below	4.54 ± 3.82	21.56 (<0.001)
	Above primary school	2.42 ± 2.91	
Comorbidities			
Hypertension	Yes	2.58 ± 3.25	12.56 (<0.001)
	No	4.22 ± 3.56	
Diabetes	Yes	3.16 ± 3.46	1.72 (0.191)
	No	3.87 ± 3.54	
Stroke history	Yes	5.55 ± 3.41	−0.94 (0.347)
	No	6.43 ± 3.01	
Heart disease	Yes	3.23 ± 3.47	2.62 (0.107)
	No	4.61 ± 3.57	

* SD = Standard deviation.

**Table 4 healthcare-13-00674-t004:** Correlations among daily lifestyle habits, autonomic nervous functions, and orthostatic hypotension.

Variables	1	2	3	4	5	6	7	8
1. Nutritional status	1							
2. Poor sleep quality	−0.30 ***	1						
3. Physical activity	0.15 *	−0.31 ***	1					
4. Depression	−0.29 ***	0.40 ***	−0.31 ***	1				
5. Stress	−0.41 ***	0.49 ***	−0.28 ***	0.48 ***	1			
6. Epinephrine	0.05	−0.12	0.13	−0.17 *	−0.04	1		
7. Norepinephrine	−0.13	0.16 *	−0.18 **	−0.13	0.08	0.36 ***	1	
8. Orthostatic hypotension	−0.36 ***	0.49 ***	−0.39 ***	0.48 ***	0.50 ***	−0.12	0.12	1

* *p* < 0.05, ** *p* < 0.01, *** *p* < 0.001.

**Table 5 healthcare-13-00674-t005:** Determinant variables for non-neurogenic orthostatic hypotension.

Variables						
		B	SE	β	t	*p*
(Constant)		1.55	3.63		0.43	0.670
Age Under 75 years (vs. 76 years or older)		−0.75	0.39	−0.11	−1.92	0.055
Education level Primary school or below (vs. above primary school)		0.55	0.41	0.08	1.36	0.176
Daily lifestyle habits	Nutritional status	−0.20	0.10	−0.12	−2.06	0.040
	Poor sleep quality	0.15	0.05	0.18	2.83	0.005
	Physical activity	−0.01	0.00	−0.13	−2.20	0.032
	Depression	0.23	0.07	0.22	3.41	0.001
	Stress	0.04	0.02	0.18	2.61	0.001
Autonomic nervous functions	Epinephrine	−1.02	1.32	−0.05	−0.77	0.442
	Norepinephrine	1.01	1.21	0.05	0.84	0.405
Adjusted R^2^ = 0.405, F (*p*) = 17.19 (<0.001), VIF = 1.147~1.611						

SD = Standard deviation.

## Data Availability

The raw data supporting the conclusions of this article will be made available by the authors upon request. Requests to access the datasets should be directed to HKO, katie5@daegu.ac.kr.

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
