# Peer review of "Effects of Daily Lifestyle Habits on Non-Neurogenic Orthostatic Hypotension in Older Adults in South Korea: A Cross-Sectional Study"

_healthcare, 2025, doi:10.3390/healthcare13060674_

Round 1
Reviewer 1 Report
Comments and Suggestions for Authors
- inclusion and exclusion criteria should be better described.
- a flow-chart of the study should be included.
- authors should better describe their population. It is difficult to identify the daily pharmacological background of the patients. Drugs might impact on the occurrence of OH. Please discuss such a point.
- authors evaluated the determinants of the OH. Indeed, no multivariate regression analysis was performed. How did confounding factors impact on final outcomes? please provide.
- Congestive status also influences the OH. Please discuss such a point in relation to the paper from J Cardiol. 2020 Jan;75(1):47-52.
Reviewer 2 Report
Comments and Suggestions for Authors
This paper presents interesting findings regarding lifestyle habits such as qualities of sleep, physical activity, and psychological factors that may contribute to orthostatic hypotension in the elderly. The study was well designed, and the measurements were adequately described. The sample size was small; however, the findings are significant and contribute a new approach to understanding the etiology of orthostatic hypotension, particularly in the elderly.
Reviewer 3 Report
Comments and Suggestions for Authors
This cross-sectional study examined the association between daily lifestyle habits and non-neurogenic OH in 217 older adults in South Korea. They found that lifestyle habits (nutrition, sleep, physical activity, and psychological status) were important factors, while ANS function had no effect on nonneurogenic OH in older adult. There are several main concerns that author should solve.
1 Over a half of cited references did not come from recent publications(within the last 5 years), the authors may be suggested to update some cited references.
2 In Materials and Methods section, the detection description of plasma epinephrine and norepinephrine levels were simple, please provide detailed method.
3 In Materials and Methods section, the authors described that “thus, patients diagnosed with Parkinson’ disease, diabetes mellitus, chronic renal failure, or any other known neurodegenerative diseases were excluded”. However, in Table 3. there are comorbidities variables, please explain these contradictory results.
4 Why only age and years of education were used as the control variables?
5 In the legend of Table 5, the author should be suggested to delete “ * SD=Standard deviation”, and added the legend of abbreviated symbols.
6 The authors focused on nonneurogenic OH associated with aging and excluded neurogenic OH due to underlying autonomic dysfunction in the recruited participation. They detected plasma catecholamines levels (autonomic nerve system function) and analyzed their association with daily lifestyle habits and non-neurogenic OH. It seems unnecessary to analyze the association between autonomic nerve system function and nonneurogenic OH in the recruited participation.
Round 2
Reviewer 1 Report
Comments and Suggestions for Authors
authors well addressed previous comments. The paper improved very much
Reviewer 3 Report
Comments and Suggestions for Authors
The authors have solved my concerns.